# The Effect of Mining Activities on the Paleokarstic Features, Recent Karstic Features, and Karst Water of the Bakony Region (Hungary)

**Márton Veress**

Department of Geography, Eötvös Lóránd University, Savaria University Centre, 9700 Szombathely, Hungary; veress.marton@sek.elte.hu

**Abstract:** This study describes the direct and indirect effects of mining on the karst of the Bakony Region. For this, the results of geological and mining research of the last century, the results of hydrological research of fifty years, as well as the investigations of several decades on the karst of the mountain region are used. Direct effects include the exploitation of filling materials (limonite, kaolinite, manganese ore, and bauxite) from paleokarst features, dolomite rubble, activities exploring or destroying cavities, and the pollution of cavity systems with mining waste (dirt). An indirect effect is karst water extraction. Mining activities (coal and quarrying) resulting in the development of pseudokarstic features are also mentioned here. It can be stated that the effects on the karst and karst features may be permanent and even renewing, but the original state may also have returned or can be expected in the near future. Damages may be local or regional. A regional effect is the decrease in karst water level, which has the most significant effect on the environment, but it has already reached its original state by now.

**Keywords:** Bakony region; mining; karst; karst depression; pseudokarst; karst water

## 1. Introduction

This study describes the effect of mining on the karst of the Bakony Region, which is the southwestern part of the Transdanubian Mountains. It is surrounded by the Little Hungarian Plain in NW, by Lake Balaton in SE, and the micro-regions of the Balaton Basin (Balaton Riviera, Tapolca Basin), the Great Hungarian Plain (Mezőföld), and the Vértes Mountains in NE (and Mór Graben with graben structure). Its elevation is 150–700 m and its area is 4300 km². Its largest area is the Bakony Mountains (2200 km²). Parts of the Bakony Region are the Keszthely Mountains, Northern Bakony, Southern Bakony, Balaton Uplands and Bakonyalja (Figure 1).

The mountains are of a block structure, where more elevated blocks are mountains, and less elevated blocks are basins. They are mainly built up of Triassic dolomite, which is significantly widespread on the surface too [1]. However, Triassic, Dachstein, and Jurassic limestones frequently occur in great expansion on blocks of higher elevation. Cretaceous and Eocene limestones of small thickness can mainly be found in a larger and smaller expansion on blocks with medium elevation, mostly covered with loess. Blocks of lower elevation (basins) are mostly covered by non-karstic rocks (Middle Oligocene–Lower Miocene gravel, Figure 2). In the mountains, rocks of various ages constitute bands of NE-SW direction. Perpendicular to this, the Bakony Region is of asymmetrical structure. In the SE, older Palaeozoic metamorphic rocks are exposed to the surface. In the middle part, the Triassic floor constitutes a synclinal of NE-SW direction where transgressions (often archipelagic) developed. Its north-western flank is incomplete: although Triassic rocks are exposed, the Palaeozoic features subsided along stepped faults and constitute the floor of the basin of the Little Hungarian Plain [1].

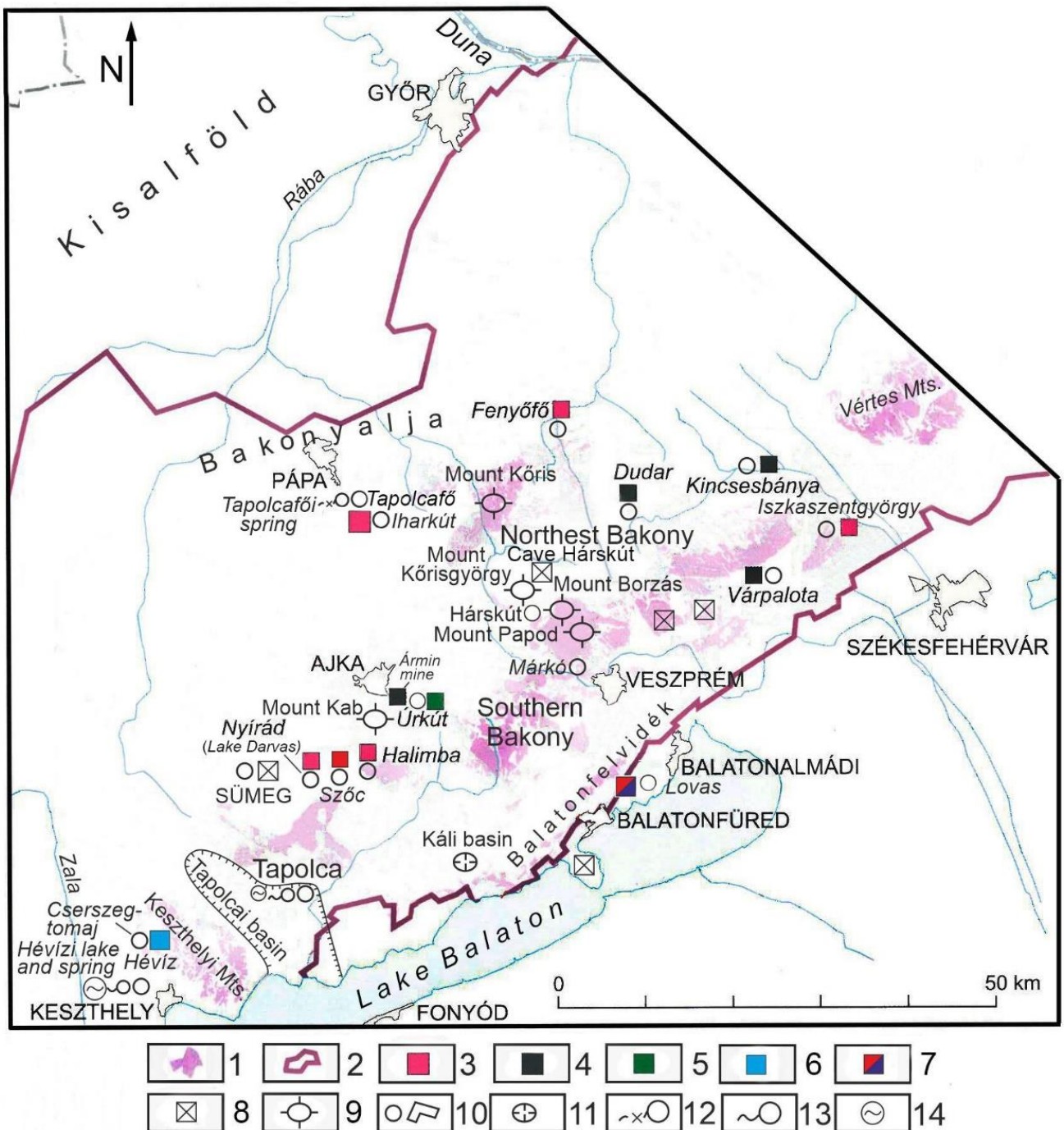

**Figure 1.** The Bakony Region and main mining sites are mentioned in this study (own figure). Legend: 1. infiltration sites, 2. border of karst water storage, 3. bauxite mine, 4. coal mine, 5. manganese mine, 6. kaolin mine, 7. paint mine, 8. stone and rubble mine, 9. mountain, 10. settlement, 11. basin, 12. karst spring that went dry, 13. active karst spring, 14. spring lake.

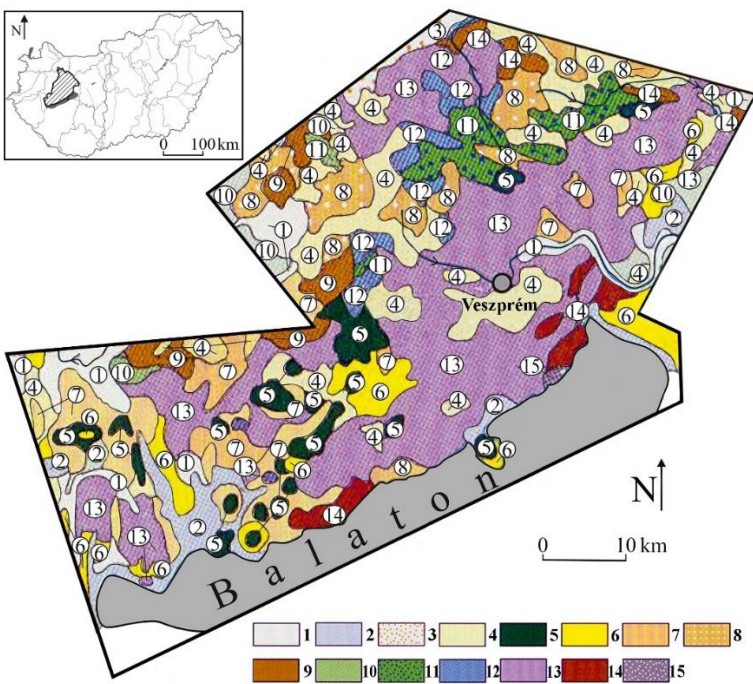

**Figure 2.** Geology of the Bakony Mountains [2]. Legend: 1. fluvial sediment, 2. Holocene torf, 3. Holocene wind-blown sand, 4. Pleistocene loess, 5. Pliocene basalt, 6. Upper-Miocene freshwater limestone, 7. Middle Miocene limestone, 8. Oligocene gravel, 9. Eocene limestone, 10. Upper Cretaceous limestone, 11. Lower Cretaceous limestone, 12. Jurassic limestone, 13. Triassic limestone, dolomite, 14. Permian sandstone, 15. old Paleozoic phyllite.

Karst develops on soluble rocks (limestone, dolomite, and evaporite). The infiltrating water with carbonic acid creates cavities (caves), fills them, and flows towards the mountain margin (karst water), where it emerges in springs. The infiltrated water constitutes a three-dimensional system whose surface is the karst water level, which undergoes fluctuation of various degrees and periods due to natural and artificial effects, at another time a one-way rise or subsidence [3,4]. Since karst rocks drain water, neither a surface water network nor valleys develop (they are only formed at sites where the valley is inherited from the non-karstic cover or when the karst water level is situated at the valley floor). The dissolved material is transported into the karst with the infiltrating waters; therefore, surface karst features are closed. These are karren, dolines, ponors with blind valleys, and poljes. The material transported in the solution precipitates as freshwater limestone.

Features resembling karst features can also develop during non-karstic processes (e.g., caves and surface depressions). These are pseudokarst features that may also be formed due to mining. Features developing at present are recent karst features, while those that developed earlier and not forming now are paleokarst.

The paleokarst of the Bakony Region is borne by the peneplain that developed during tropical karstification [5]. Its paleokarst is mainly of Cretaceous age [6,7], but others may also occur [6]. The surface karst features mostly developed on Triassic dolomite; the cover is most frequently Eocene limestone and Middle Oligocene–Lower Miocene gravel [6]. During karstification karst depressions (solution dolines), poljes and inselberg karst developed [5,8,9]. Its present surface karst is of an island-like pattern and covered karst that developed on blocks of medium elevation [10]. Features of its covered karst are suffosion dolines, but solution dolines also occur in small numbers at the margin of the mountains [10].

The effects of mining on the karst and karst features are overviewed and classified. This complex overview may have importance in education and environmental protection. The here described approaches might be applied to the qualification of other karst areas.

## 2. Direct Mining Effects on the Karst

In case of direct effect, the mining activity affects the karst feature: exploitation of minerals or supplying the karst system with materials. The first is some kind of erosion; the latter is accumulation. These are primary effects, but secondary effects may also occur in both cases; for example, during exploitation, the pollution takes place through fuel infiltration.

Direct effects involve the exploitation of the filling material and dolomite rubble, the exploration of karst cavities, the transportation of the mining waste into the karst, as well as feature development and destruction due to mining.

The original feature became exhumed during surface mining when the filling material that accumulated in karst depressions acting as traps was exploited, and the cover was removed. If the mineral resources to be exploited were at a greater depth, exploitation would occur through underground mining operations instead of open-pit mining. The strip pit is either constituted by depressions, or they are parts of the strip pit. Usually, remnants of the mined filling material can still be found in depressions. The original karst feature does not only differ from the original but also because its slope was transformed due to subsequent tectonic stress, and larger and smaller sections of the bearing rock may have been removed.

The exploited material can be material containing limonite, hematite, kaolin (kaolinitic clay), manganese ore, and bauxite.

The limonite–hematite material was exploited from two Palaeolithic so-called paint mines between the settlements of Lovas and Felsőőrs in the Balaton Uplands. The paint mines were detected during dolomite rubble mining [11]. According to artifacts found here, they originate from the end of the first third of Würmian $\frac{1}{2}$ interstadial [11,12]. Based on the age of the mining of the paint mine; as a mine, it is probably unique. Unfortunately, its exact site is unknown; thus, geomorphological conditions cannot be studied subsequently. According to published figures, the mined material may have accumulated in the karstic bedrock of the dolomite, or it may have been mixed with this material during rubble formation.

Furthermore, 1–2 km away from Cserszegtomaj (Keszthely Mountains), kaolinitic clay occurrence can be found partly in superficial outcrop and partly covered thinly [13], which are the fills of karst depressions. In his later work, [14] mentions the filling material as bauxite material. The fillings were exploited by hand; thus, deep dolines of 10–50 m with vertical walls were exposed in the Triassic dolomite [8,13]. According to Bárdossy [13], the development age of doline development coincides with the age of Nyirád-Halimba bauxite deposits which is regarded as Lower Cretaceous by [6].

There are several manganese ore occurrences in the mountains. The oxidized manganese ore entered into the karst depressions of Úrkút (Csárda-hegy) by reworking; its cover is of Lower and Middle Eocene, while the bearing karst features deepen into Lower Liassic (Jurassic) limestone [15]. The manganese ore was exploited by hand in the twenties of the 20th century (here, there is a manganese ore mine with an underground mining operation where exploitation still takes place). As a result, a tropical-type doline with steep slopes was exposed with many solution features on its slope, a gentler, rather temperate solution doline, and a terrain with pinnacle karren features (Figure 3).

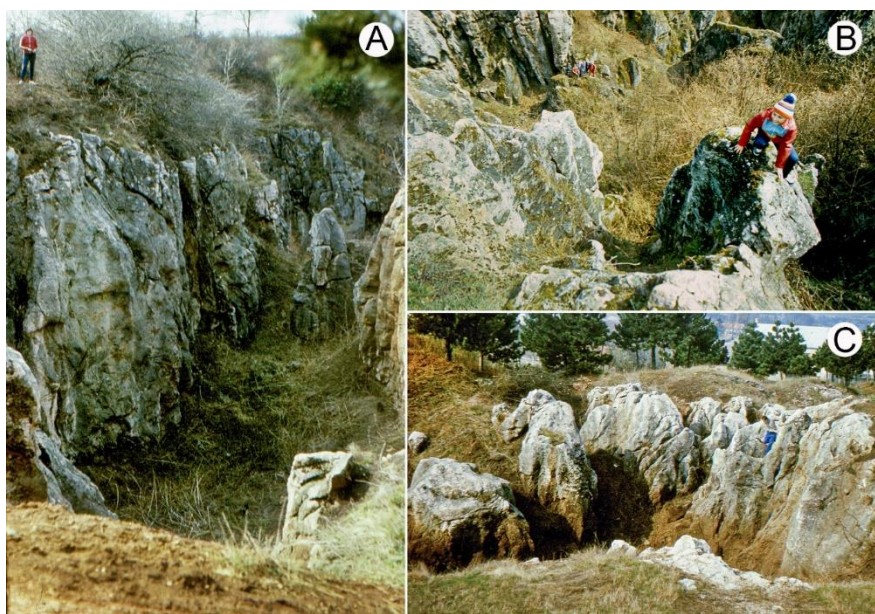

**Figure 3.** Exposed paleokarst features of Csárda-hegy (hill) (own figure). Legend: (**A**) tropical doline with steep slopes: (**B**) solution pit of doline slope; (**C**) temperate doline with gentle slopes and pinnacle karren.

Bauxite is widespread in the mountains and may constitute a lenticular colony (diameter is about 20 m, thickness is 30–50 m, the bearing features is shallow doline), a deep-doline colony (diameter is 30–50 m, thickness is 80–100 m, the bearing features are deep doline), a bed-like colony (longitudinal expansion is several hundred meters, the bearing feature is polje or uvala), a structural colony (the bearing feature is tectonic graben), a deep-doline and structural colony (a combination of karst feature and structural feature), and a canyon-like colony when it is several hundred meters long; the karst feature is a blind valley [6,14,16]. Under tropical climates, bauxite is the weathering residue of the rocks of the non-karstic terrain [16], which was transported into the karst depressions, and then accumulated in them but may have formed in situ as well [17], or it may have been transported from below by transgression [18]. The products of lateritic chemical weathering may originate from the peneplain built up of metamorphic rocks surrounding the mountains from S and SE [19]. It cannot have formed in situ since it is separated from the uneven bedrock without transition; however, the iron crust is mostly present on the bedrock, which may have formed during and after bauxite accumulation [16].

The bedrock of karst depressions is Triassic Main Dolomite and Dachstein limestone [6,18], but the material of the depressions of Triassic Main Dolomite may have been reworked and accumulated in Late Cretaceous depressions [6]. Hungarian bauxites—the bauxites of the Bakony Region—can be classified as belonging to various development stages. The second stage (the first stage occurs in the Villány Mountains) involves the colonies of the Albian Lower Perepuszta (near the settlement of Zirc) and those of Bakonyoszlop. Their bedrock is Upper Triassic Dachstein limestone, and their cover is Upper Cretaceous clay marl. The third stage is the Upper Cretaceous Senonian stage. This includes the colonies of Halimba, Iharkút, and Csabpuszta (Sümeg). Their bedrock is Upper Triassic Main Dolomite, and their cover is Upper Cretaceous limestone or fluvial sandstone, clay, and clay marl. The age of the fourth stage is at the Cretaceous-Eocene boundary. The colonies of Fenyőfő, Dudar, Bakonyoszlop, Szőc, Halimba, and Nyírád belong to this. Their bedrock is Upper Triassic Dolomite, and their cover is Eocene limestone [14,16]. However, there are colonies whose cover is Eocene limestone or Middle Oligocene-Lower Miocene gravel and Pleistocene sediment [6,16]. If the caprocks were thinner, their exploitation took place by open-pit mining (Iharkút, Nyírád, Szőc); if they were thicker, they were exploited by underground

mining operations (Iszkaszentgyörgy, Halimba, Fenyőfő, Figure 4). Karst depressions, or the terrain dissected by depressions, became exhumed during open-pit mining (Figure 5).

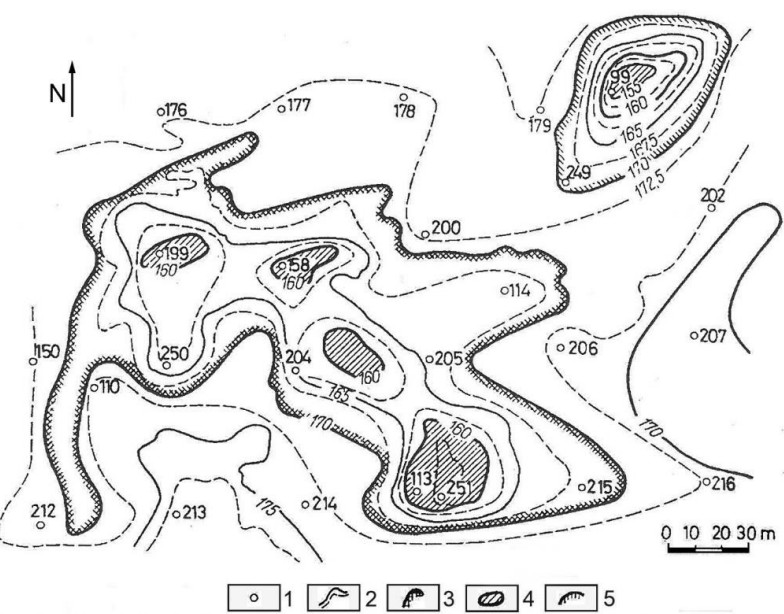

**Figure 4.** Bedrock map of the Upper Triassic Main Dolomite of the area of Halimba [20]. Legend: 1. drilling and its number, 2. contour line, 3. uvala, 4. the lowest point of doline, 5. doline.

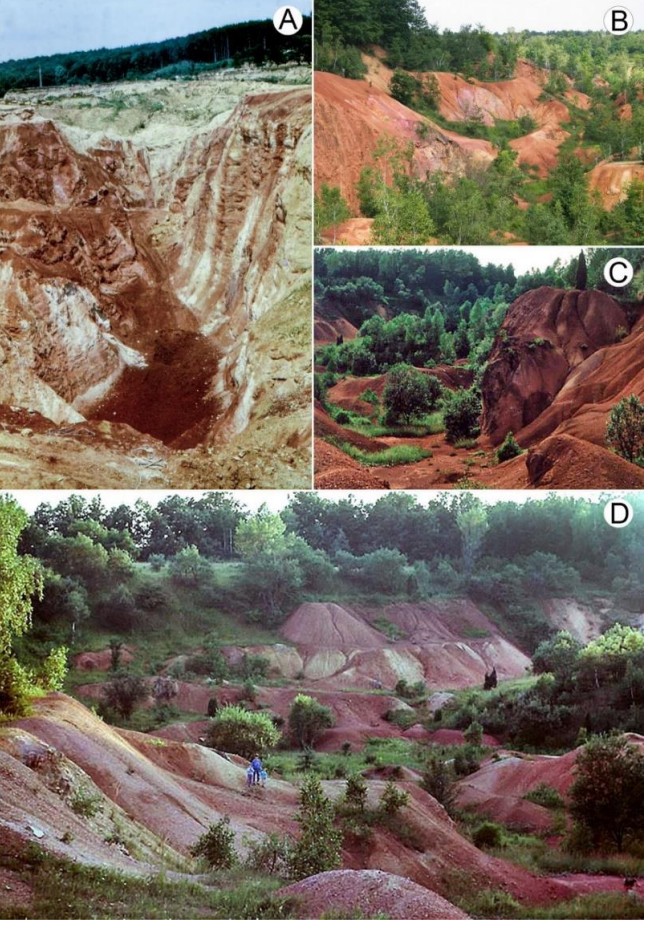

**Figure 5.** Surface depressions exhumed during bauxite mining that developed Upper Triassic Main Dolomite: (**A**) Iharkút, (**B**–**D**) Lake Darvas (own figure).

The separation of the manganese ore of Úrkút from mining waste was carried out in the manganese washing plant. The so-formed manganese mud led into one of the depressions of Kab Mountain. The sludge appeared in the spring of the valley located 1.5 km away from this site about three weeks later [21]. The sludge was led into the doline at a rate of 1 $m^3$/min, which had an annual quantity of 0.5 million $km^3$ [21], although the duration of supply is unknown.

Dolomite rubble beds are mined on dolomite at several places between the settlement of Márkó and Várpalota in the mountains. The thickness of the rubble ranges from some meters to several ten meters. According to [20], its thickness may also exceed 50 m at Nyirád. The reason for rubble formation is that the calcareous material sticking to dolomite crystals is dissolved, and the dolomite falls into parts [3]. At sites of rubble exploitation, the wavy surface of lower karstic bedrock dolomite is exposed. Inside strip pits, where the rock underwent rubble formation to a lesser degree, the rock was left, and thus, the strip pits are dissected by mounds.

Several caves have been exposed during limestone mining. The longest is the recently discovered several km-long Hárskút Cave, which opens in the quarry of the Kőrisgyörgy Hill. The opened-up caves do not exist for a long time; they become shorter and then decay (e.g., in the quarry of Sümeg).

There was mining activity on the sandstone fields of the Kál Basin. The Pannonian sand became cemented with the amorphous silica that had precipitated from the hot water of post-volcanic activity [22] on which a rich karren landscape developed [23]. In the area of the Szentbékálla block field, the material of blocks was partly exploited (during this, several karren features may have been destroyed) for millstone, while that of Kővágóőrs block field was used to build houses. Similarly, quarrying damaged the (calcareous, quartzic) thermal cones of the Tihany Peninsula. Their rocks were used for construction work. Larger and smaller quarries were created on their sides, but at others, almost half of the cone material was exploited, and thus, the cavity of the inner side of the cone was exposed [24].

## 3. Indirect Effect on the Karst

An indirect effect develops when the anthropogenic activity does not affect the karst feature, but the alternation of the hydrology of the karst system. The consequences of the changes in hydrological conditions are regional, diverse, and occur fast. The uniform karst water system of the mountains is the main karst water, which primarily developed in the Triassic Main Dolomite. According to drilling and spring data, its former surface was at an elevation of 117.5–154.2 m at the margin of the mountains; it was of an increasingly larger altitude towards the center of the mountains. With the help of the data, a karst water level map was made, which was updated annually. Karst water observation wells were created in order to monitor water level changes; the number of these wells reached 200 within some decades. According to the spring data, the karst water level reached an elevation of 240 m in the inner part of the mountains [25], while it was at an elevation of 280 m between the settlements of Szentgál and Nagyvázsony based on the data of the constructed map of the karst water level [26]. In the latter area, the karst water swelled back because of the impermeable rocks of Southern Bakony. In the interest of the safe exploitation of mineral resources (without karst water flood), an artificial lowering of the karst water level was started in the environs of the mines with pumping. Its intensity reached a value of 460 $m^3$/min. The water level sank below the mines and reached 100 m at some sites [26]. Depressions developed on its surface. Not only the depth, but also the expansion of depressions increased, and adjacent depressions coalesced. Maximum values of water level subsidence are shown in Figure 6.

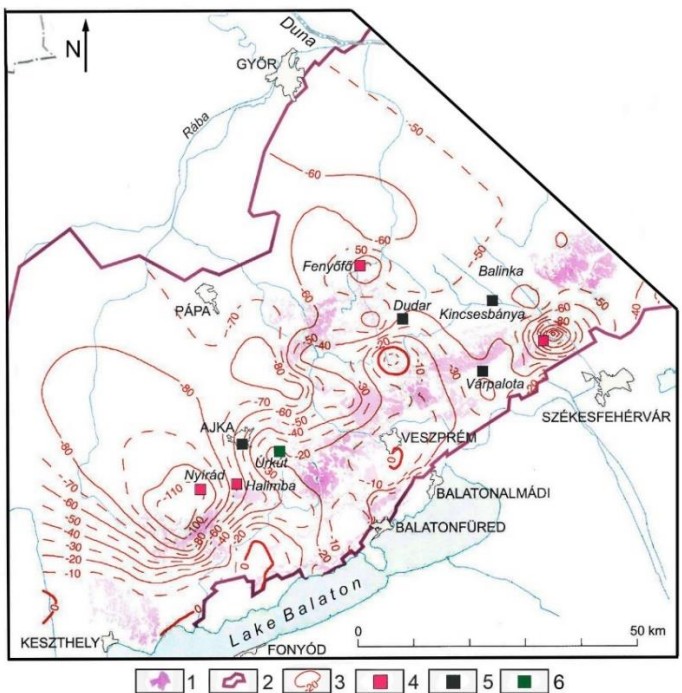

**Figure 6.** Artificial water level lowering in the Bakony Region [26]. Legend: 1. infiltration areas, 2. limit of karst water storage, 3. artificial karst water level lowering (m), 4. bauxite mine, 5. coal mine, 6. manganese mine.

## 4. The Artificial Water Level Lowering Resulted in the following Facts

(1)     Water level subsidence reached the margin of the mountains. At sites where the level was below the level of karst springs, the discharge of springs decreased (Spring lake of Hévíz), or they went dry (Tapolcafő springs). At the town of Tapolca, not only the spring of Lake Malom (Figure 7) went dry, but the cave providing this site with water went dry too.

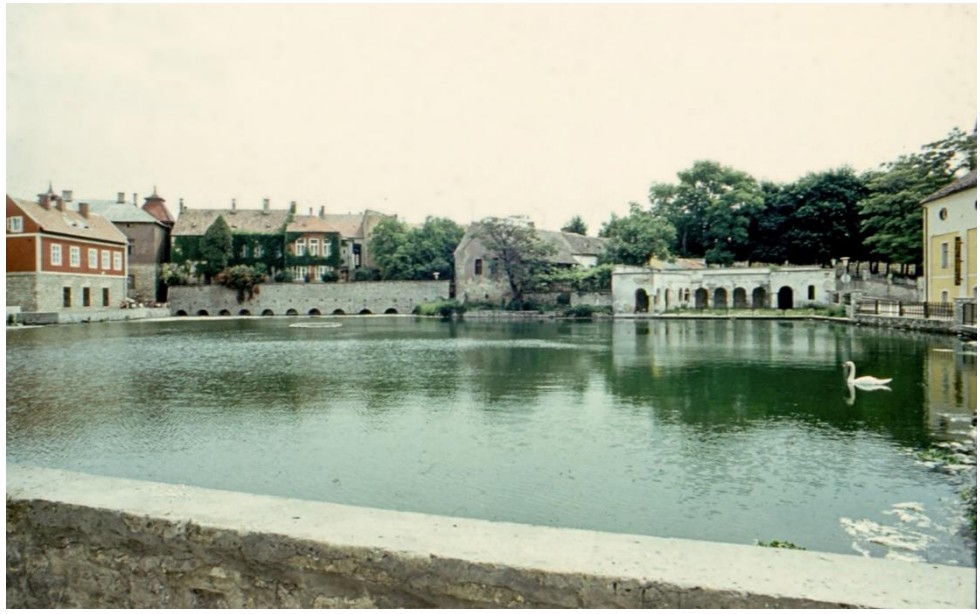

**Figure 7.** Lake Malom of Tapolca which developed by the damming of a stream, but it is also supplied by luke-warm karst springs (therefore, the water temperature is 18–20 °C) (the spring water originates from Sarmatian limestone) (own figure).

(2)     Since the karst waters of the mountains flowed into the karstic floor of the surrounding basin (Little Hungarian Plain) and into its sediments, water level lowering could also be observed in the Little Hungarian Plain (Figure 6), and the direction of water flow partly changed (at some places water flow oriented towards the mountains).

## 5. Anthropogenic Pseudokarst Features Created through Mining

Several coal mines were developed in the area of the coal basin at Ajka. One of them is the Ármin Mine of Bocskor Hill. In the mine, there are seven coal beds below each other in Upper Cretaceous limestone [27]. One of them is shown in Figure 8. Closed depressions were formed at the surface above the mine. Their number exceeds 100, and they are developing at present too. They are areic, aligned in rows, elongated, and grike-like, but there are also features with circular ground plans [23] (Figure 9). Where cover is present, they are inherited onto the cover, and they resemble subsidence dolines (pseudokarstic subsidence doline) in this case.

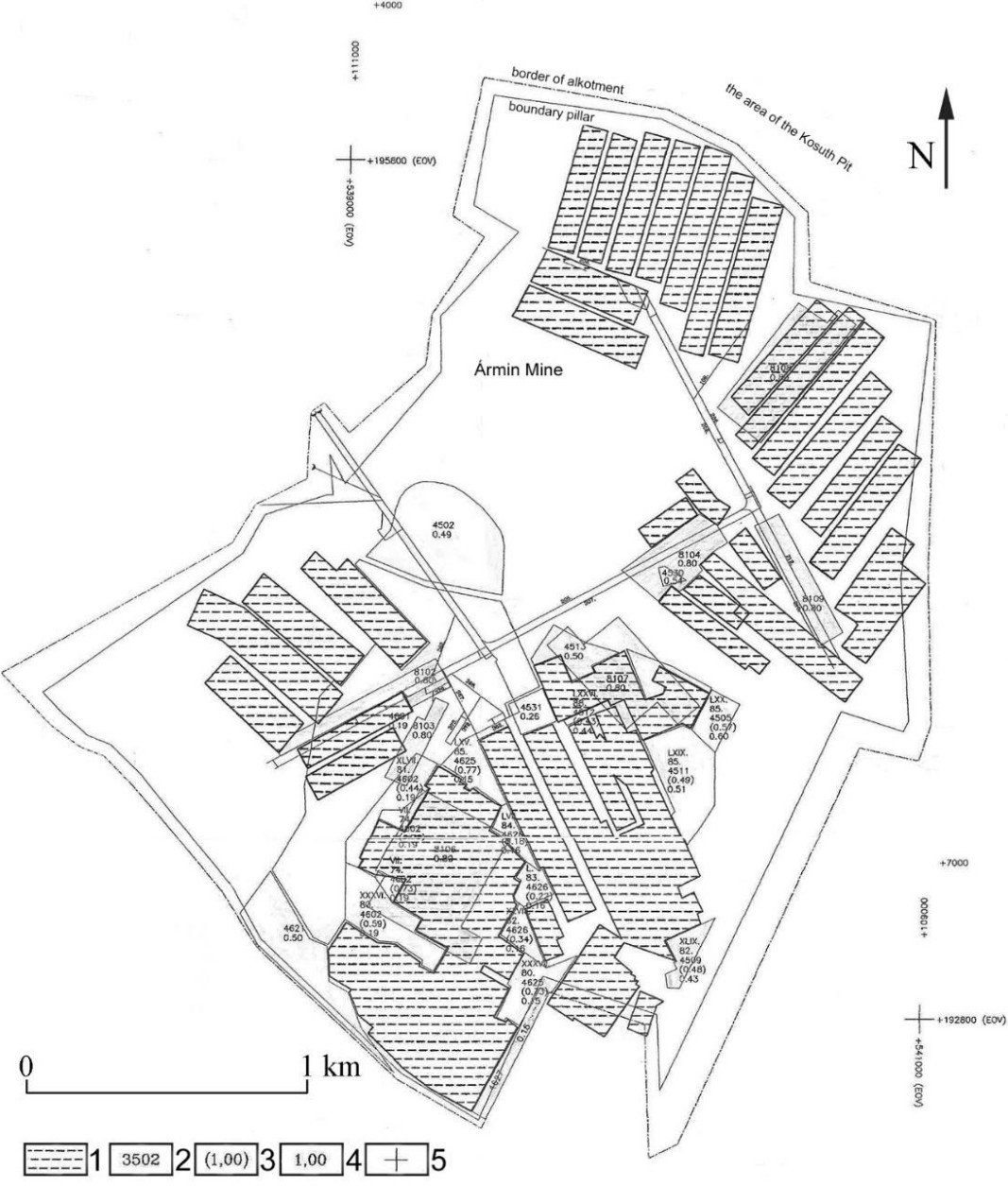

**Figure 8.** Map of bed IV of Ármin mine [28]. Legend: 1. excavation, 2. block number, 3. abandoned mineral reserves, original mineable value 4. mineable value, 5. map coordinate.

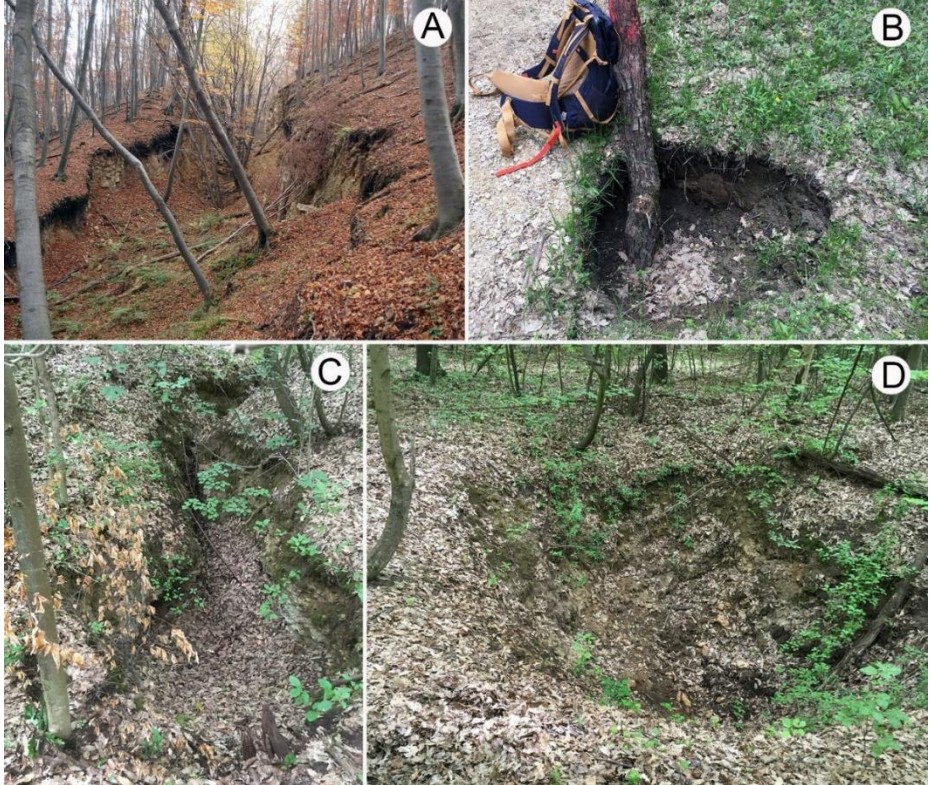

**Figure 9.** Pseudokarst features from Bocskor Hill: (**A**) graben feature (only partly areic) (Kopecskó); (**B**) circular pseudokarst dropout doline (own figure); (**C**) pseudokarst dropout doline of grike pattern (feature with collapse) (own figure); (**D**) circular pseudokarst suffosion doline (own figure) (the features developed on Middle Eocene limestone).

Their development can be explained by the material equilibrium resulting from the mining of coal beds situated below each other. During this, the covering Eocene limestone warped, tension stress developed at the margin of the warping, which resulted in the development of tension grikes in the rock, then the cover collapsed into some grikes [23] (Figure 10).

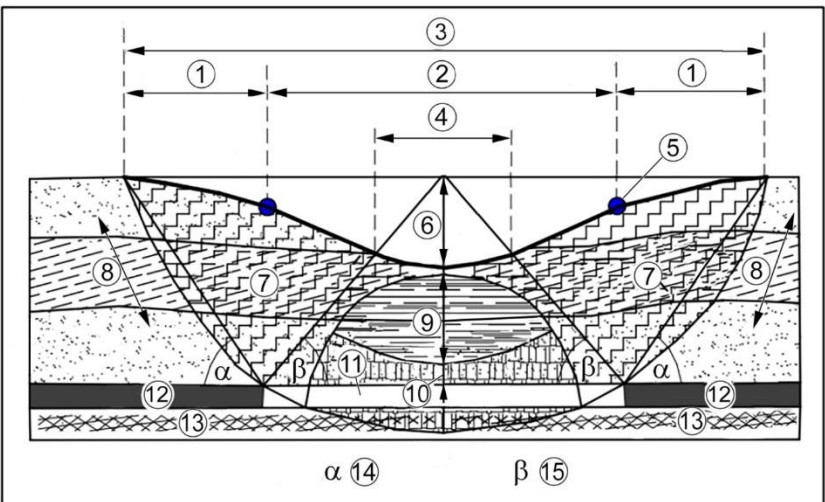

**Figure 10.** Subsidence trough developed by the sinking of the vault [29–31] (Legend: 1. tension zone, 2. warping zone, 3. original terrain, 4. sole subsidence zone, 5. inflexion point, 6. maximum subsidence, 7. deflection zone, 8. cover layers, 9. fracture zone, 10. collapse zone, 11. excavation field, 12. coal beds, 13. bedrock layers, 14. boundary angle, 15. undercut angle.

From the 18th century to the middle of the 19th century, several glasshouses were operated in various sites in the mountains [32]. For this reason, the rock was exploited at several places creating depressions with a diameter and depth of some meters necessary for the glasshouses. Their number is probably several thousand. They occur in groups (e.g., on the Kőris Hill, Borzás Hill, and Papod Hill). Glass sludge developed during smelting frequently and was found around them (the pieces are porous, and the cavernous coating of the former melt is visible on their surface).

Limestone was exploited at several sites (village, farm) for house building by hand. The exploitation scars are also closed; however, the side margins are straight in the ground plan (but the adjacent margins enclose an angle of almost 90°), their floor and slopes are covered with debris, and there is usually exploited debris around them too. These characteristics show their artificial origin.

## 6. Conclusions

The relationship between mining and karst in the Bakony Region is described. The main mining effects are classified, and we overviewed what opportunities the karst offered to mine. In the mountains, mining affected the karst of the Bakony Region directly (mining of the material of depressions and of the dolomite rubble, pollution of karst cavities by mining, cavity and cave destruction due to rock mining) and indirectly (karst water extraction). Mining also resulted in the development of pseudokarst features.

Environmental damages caused by mining can be permanent in the karst landscape (rubble mining); others can be solved partly by reconstruction work (e.g., refilling depressions whose material was exploited), but a complete regeneration can also be expected (karst water level). The majority of damage is mostly local; however, karst water extraction affected the whole mountains and their environment, but its degree was different at various sites. Fortunately, due to the reversible character of water balance, the recovery of the original state can be expected after a certain time. Signs of regeneration have been visible at several sites for the past 30 years since the 1990s (there is karst water in the Tapolca Cave again, but several springs have started to operate again).

**Funding:** This research received no external funding.

**Informed Consent Statement:** Not applicable.

**Data Availability Statement:** No new data were created or analyzed in this study. Data sharing is not applicable to this article.

**Conflicts of Interest:** The authors declare no conflict of interest.

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
