# Peer review of "The Effect of Mining Activities on the Paleokarstic Features, Recent Karstic Features, and Karst Water of the Bakony Region (Hungary)"

_mining, doi:10.3390/mining2040042_

Round 1

Reviewer 1 Report

This article is very interesting and shows very important effects on natural resources. But it does not have documented methodology and results. Paying attention to the following points will improve it. After the authors edit, I would like to review it again

1- The title or attention to the goal is not chosen correctly.

2- Results, materials and methods are not presented in the abstract.

3- Innovation should be presented in the introduction

4- The direct effects of mining have not been well explained

5- The methodology of the indirect effects section is not well explained

6- This article is more of a review article and it is better to present the methodology and written results section

Author Response

Please find my answer in the attached file, please.

Reviewer 2 Report

1- The geology of the studied area and the geological map should be presented.

2- The abstract should be re-edited and the achievements made more complete.

3- Provide more complete explanations about the karstic bauxites of the region.

4- The photos presented about the karstic phenomena of the region, in which geological formations have they been seen?

5- Apply the corrections I made to the text of the manuscript. See the attached PDF file.

Author Response

Please find my answer in the attached document, please.

Round 2

Reviewer 1 Report

This manuscript has been greatly improved after the suggested comments and is ready for publication.

Reviewer 2 Report

The respected authors have made all my corrections.